# Validation of Reference Genes in a Population of Blueberry *(Vaccinium corymbosum)* Plants Regenerated in Colchicine

**DOI:** 10.3390/plants11192645

**Published:** 2022-10-08

**Authors:** Francisca Valenzuela, Vivían D’Afonseca, Ricardo Hernández, Aleydis Gómez, Ariel D. Arencibia

**Affiliations:** 1Centro de Biotecnología en Recursos Naturales, Facultad de Ciencias Agrarias y Forestales, Universidad Católica del Maule, Ave San Miguel 3605, Talca 3466706, Chile; 2Departamento de Ciencias Preclínicas, Facultad de Medicina, Universidad Católica del Maule, Ave San Miguel 3605, Talca 3466706, Chile; 3Doctorado en Biotecnología Traslacional. Facultad de Ciencias Agrarias y Forestales, Universidad Católica del Maule, Ave San Miguel 3605, Talca 3466706, Chile

**Keywords:** reference genes, gene expression, colchicine, RT-qPCR, *Vaccinium corymbosum*

## Abstract

For the first time we report the validation of reference genes in plants from a population of blueberry (*Vaccinium corymbosum*) clones cultured in vitro on a colchicine-supplemented medium. Nodal segment explants of the cultivar Duke were regenerated by organogenesis under different periods of colchicine 1 mg/L exposure (1, 2, 3, 5, 30 days). The clones selected for the study showed variability for phenotypic traits after 2 years of adaptation to field conditions, compared to plants of the donor genotype that were regenerated on a medium without colchicine. *Vaccinium myrtillus* (*GAPDH*) and *Vaccinium macrocarpon* (*ATP1, NADH, RPOB* and *COX2*) were used as reference genomes for primer design. The results show that colchicine treatments can cause genomic changes in blueberry plants. At the molecular level, exposure of plants to colchicine in early periods could promote an increase in gene expression of specific genes such as *ATP1, COX2, GAPDH, MATK, NADH* and RPOB. However, prolonged exposure (30 days) could decrease gene expression of the genes studied. For qPCR assays, the primers designed for *ATP1, COX2, GAPDH* and *MATK* genes showed high efficiency. In addition, the *GAPDH*, *ATP1*, *NADH* and *COX2* genes showed high stability and could be recommended as potential reference genes for gene expression assays in Vaccinium.

## 1. Introduction

The production of blueberries (*Vaccinium* spp.) has increased in recent years worldwide due to their beneficial characteristics for human health, explained by their high content of secondary metabolites with antioxidant properties of high value for the functional food industry [1,2]. Chile is an important producer and exporter of blueberries in the southern hemisphere, increasing in recent years the area planted in different regions. According to estimates by the Chilean Blueberry Committee of ASOEX, exports of this fruit in the 2020–2021 season were 154,050 tonnes (fresh and frozen blueberries), with the selection of commercial varieties being one of the factors that determined the high production of blueberries [3].

The blueberry belongs to the family Ericaceae and has been classified in the subfamily Vacciniaceae, subgenus Cyanococcus, genus Vaccinium. The main commercial varieties of blueberries are highbush (*Vaccinium corymbosum L.*), lowbush (*V. angustifolium Aiton*), rabbiteye (*V. ashei Reade*) and southern highbush (interspecific hybrids of *Vaccinium*). In addition, medium-high highbush blueberry (lowbush blueberry and intercropped highbush blueberry) has entered the trade in recent years [4]. Breeding and selection of modern blueberry varieties with distinctive agro-productive traits, resistance to biotic and abiotic stresses and increased nutritional values has been an ongoing effort of traditional genetic improvement programs [4,5]. Plant genetic improvement can also be assisted by biotechnological approaches; among these, one of the most widely used is the application of colchicine treatments to induce different degrees of genetic variability resulting from polyploidy [6,7].

Polyploidy is a heritable state that maintains more than two complete sets of chromosomes and plays an important role in genetic and phenotypic diversity, plant evolution and adaptation, species diversification and breeding programs [8,9,10]. In general, polyploid plants are classified as autopolyploid, a term that refers to the repetition of the genome from similar ancestors or donors, e.g., somatic cell explants from a donor from tissue culture [11,12]. Otherwise, they are referred to as allopolyploids when the resulting plants originate from the hybridization of parents with different genomic content and the duplication of this hybridized genomic material occurs at that point [13].

Polyploidy is being studied in various contexts, including molecular biology, gene expression and genomic structure, while its functions and complex interrelationships in biological processes are still unclear. In this sense, genomics could provide an efficient platform for future studies, where interdisciplinary approaches should be crucial in identifying the functions and regulatory mechanisms of polyploidy [10].

Other research recognizes rapid and dynamic changes in genomic structure and gene expression in polyploid plants. Results show similar structures of uniparental gene regulation and non-additive gene expression in regulatory pathways such as growth, development and stresses, leading to the conclusion that epigenetic mechanisms, including chromatin modifications and small RNAs, play a key role in determining molecular and phenotypic novelty, mainly in allopolyploid plants [12].

The development of tissue culture technology has proven to be a flexible platform to support plant breeding programs. As an example, by integrating genetic engineering, mutagenesis or somaclonal variation into in vitro culture, it is possible to express, induce or evidence genetic variability as a source for the selection of elite clones for plant breeding [14]. Colchicine is an alkaloid-type metabolite obtained from meadow saffron (*Colchicum autumnale* L.) that inhibits chromosome separation during cell division, leading to chromosome duplication, and is widely used as an antimitotic agent for the induction of polyploidy [7,11,15,16,17].

For differential gene expression studies in plants, which could include populations of individuals regenerated from in vitro culture or by treatments to induce genetic variability, the quantitative PCR technique (RT-qPCR) is recommended. This approach requires accurate and reproducible measurements of transcriptional levels of target genes. However, in plants, the number of validated reference genes for this type of study remains limited. Currently, most of the normalizing genes used in plant assays are genes for fundamental cellular functions such as actin, tubulin, elongation factor 1 and 18S and 40S ribosomal RNAs [18,19,20,21]. However, it has been shown that some of them can vary in expression according to experimental conditions and depending on the tissue under study [19,22,23]. Candidate gene studies have been carried out in different species such as chicory [18], cucumber [19], potato [23], grapevine [24,25], papaya [26] and soybean [27].

For blueberry, *UBC28*, *RH8*, *CACSa* and *UBQ3b* genes showed stable expression in multiple organs in Powderblue (rabbit’s eye) and Star (southern highbush) cultivars, while *CACSa*, *RH8* and *UBC28* were the most stable genes during abscission induction experiments [28]. For Ericaceae, a set of potential reference genes (*Vc4g16320*, *CACSa*, *PPR*, *GAPDH*, *UBC9*, *Vc4g26410*, *PEX4* and *TIP41*) was also identified from public transcriptome databases, while the authors recommend the use of two or three reference genes to achieve reproducibility of results [29]. In addition, from a blueberry transcriptome database, the reference genes EF1∞, EIF and TBP showed high stability at different developmental stages under abiotic stress [30]. In a more recent study in diploid (*V. myrtilloides*) and tetraploid (*V. angustifolium f. nigrum*) blueberry floral tissues challenged with the pathogen *Monilinia vaccinii-corymbosi* (Reade) Miel (*M.vc*) and measuring pathogenesis related to the protein gene *PR3*, the genes *UBC9* and *GAPDH* were the most stable as reference genes, while *RH8* and *PPR* were the least stable [31].

In this work, for the first time, a population of blueberry plants regenerated in vitro in different colchicine treatments is used as genetic material for the verification of reference genes. The donor cultivar was Duke, while the plants selected for RT-qPCR analysis were those that showed phenotypic variability for agrobotanical traits under field conditions for two years. For gene selection, both *Vaccinium myrtillus* and *Vaccinium macrocarpon* were the genomes initially screened. In this work, the *GAPDH*, *ATP1*, *NADH* and *COX2* genes tested with high stability were validated and could be used as reference genes in gene expression assays for blueberry plants or other Vaccinium species.

## 2. Results

Of the total number of plants regenerated in colchicine treatments, 11,565 were adapted in the greenhouse, of which 8219 survived. Of these, 817 of the colchicine-treated clones from different treatments were selected for their vigor to be transplanted to the field. Both acclimatization and field phases were carried out in San Antonio de Encina, Linares. It was observed that the clones of the treatments with a longer exposure time to colchicine showed a greater dispersion in fertility (presence or absence of fruits), as well as correlating the basal diameters of the treated plants with their respective stem heights. During the first 6 months of field adaptation, agromorphological traits were evaluated, showing genetic variability among the clones under study (Figure 1). 

An example is shown in Figure 2 where variations in the shape, size and coloring of the leaves of different randomly selected clones with respect to the donor genotype are observed. Finally, a total of 38 clones showing phenotypic variability during two years were selected for the reference gene characterization study.

### 2.1. ATP1, COX2, GAPDH and MATK Gene Primers Showed Higher Efficiency in qPCR Assays

Of the 38 clones selected, a total of 26 clones showed genomic material of appropriate quality for this study. Samples with RNA concentration and purity (260/280 nm ratio between 1.5 and 2.0) were used to synthesize cDNA (Table 1). For cDNA viability, a control sample was used for all primers: *ATP1 (181 bp), COX2 (181 bp), GAPDH (174 bp), MATK (195 bp), NADH (170 bp)* and *RPOB (199 bp).* All primers were able to amplify the fragments of interest corresponding to their respective molecular weights. In addition, the viability of each cDNA was tested using the *GAPDH* gene, a gene used as a reference gene (Table 2).

For the efficiency test, a preliminary qPCR was performed for each primer set and a control sample was used in triplicate. The best qPCR assay result obtained for each primer set was used for serial dilution from 10^−1^ to 10^−6^. All sets were able to amplify a single amplicon, which can be seen in all unique peak shapes generated for each assay. The reactions generated a single amplicon, and no non-specificity was verified according to the melting curve plots (Figure 3). All primers presented an amplification curve, where amplifications were observed to advance as dilutions progressed, showing an increase of three to five cycles and affecting the threshold cycle (TC) at each dilution. Additionally, the dilution points (standard curves) of the qPCR assays can be seen in Figure 4 representing the efficiency of each primer during the dilution processes. Six dilution points were used, each point in triplicate for each gene. Almost all genes show an adequate standard curve with all six dilution points. However, the primer for the *RPOB* gene used five dilution points, and the primers for the *NADH* and *RPOB* genes have not presented all dilution points in a sloping trend (see Figure 4e,f). These results may influence the efficiency of the primers. The *ATP1*, *COX2*, *GAPDH* and *MATK* gene primers have shown an efficiency of around 91.09 to 109.36 and an R^2^ coefficient of 0.985 to 0.995. The primers with the best performance were those of *MATK* with an efficiency of 100.08 and R^2^ coefficient of 0.995, followed by the *COX2* gene primer with an efficiency of 99.02 and R^2^ coefficient of 0.994.

The primer sets for NADH and RPOB performed the lowest in this experiment with efficiency values of 113.47 and 109.73 and R^2^ coefficients of 0.954 and 0.980, respectively. The results are summarized in Table 1.

### 2.2. Gene Expression Behavior in Colchicine-Regenerated Blueberry Clones

The expression of each gene studied in blueberry plants was evaluated, showing a variable pattern with respect to colchicine treatment (Figure 5). When absolute expression was considered, the *ATP1* gene showed a slight increase in plants regenerated after 2 days of colchicine treatment; however, its expression decreased in plants regenerated after 3, 5 and 30 days of treatment (Figure 5a). For the *COX2* gene, samples from 2- and 5-day treatments show a higher absolute expression than the control; however, the median of all gene expression levels relative to colchicine treatments is lower than that of the control samples (Figure 5b). In the case of the *GAPDH* gene, colchicine treatment altered expression in all assays compared to control samples, resulting in increased expression mainly in plants regenerated in the 2 and 5 days of treatments (Figure 5c). *MATK* and *NADH* genes showed increased expression in plants from the 1-, 2- and 5-day colchicine treatments compared to control samples and the other two treatments (3 and 30 days). *NADH* showed a significant increase in expression in clones regenerated after 5 days under colchicine conditions (Figure 5d,e). Finally, the *RPOB* gene showed a slight increase in expression in clones treated for 1 and 2 days (Figure 5f). However, clones regenerated after 3 and 30 days resulted in a decrease in gene expression in all the genes studied. Treatments for 2 and 5 days in colchicine were shown to influence the expression patterns of almost all the genes studied.

To evaluate the relative expression of these genes, the gene encoding the protein *glyceraldehyde-3-phosphate dehydrogenase* (*GAPDH*) was chosen as it is widely reported for this purpose. For this purpose, the *Delta-Delta-CT* methodology was used, where all expression levels of the samples of interest were compared with the expression levels of the control samples and the *GADPH* reference gene. Furthermore, the tests were performed in triplicate, and each triplicate value obtained was only considered if the standard deviation of the obtained TCs did not exceed 0.5. The averages of the relative expression levels (major and minor) obtained for each gene in each sample of colchicine treatment (1 day, 2 days, 3 days, 5 days and 30 days) and control using the *Delta-Delta-CT* method are represented in Figure 6. The expression of the *ATP1* gene showed a slight increase in the plants receiving 2 days of treatment and an expressive increase in the clones treated with the 5-day colchicine treatment (Figure 6a). Clones regenerated in colchicine (30 days) showed the lowest expression values within the genes studied, with the exception of the *COX2* gene. In addition, the *COX2* gene expression of the control samples was higher than that of the other colchicine treatment samples (Figure 6b). In addition, this gene showed increased expression at 3 days. The *MATK* gene showed increased expression in plants with 1-day colchicine treatment, and the *NADH* gene showed increased expression in the 3-day treatment (Figure 6c,d). Finally, the *RPOB* gene had its expression significantly increased in the regenerated clones after 1 and 2 days of treatment (Figure 6e). Considering the relative expression, colchicine treatment generally promoted a decrease in expression in the 1-, 2- and 3-day treatment plants.

### 2.3. Are These Genes Stable for qPCR Assays?

Gene stability was studied using the RefFinder software. This software uses the scores generated in several programs (NormFinder, GeNorm and BestKeeper) that work on gene stability analysis for reference genes. From each program, RefFinder assigns an appropriate weight (score) to an individual gene and calculates the geometric mean of its weights for the final overall ranking. Figure 7 summarizes the stability studies of the reference genes analyzed in all samples of each colchicine treatment together (1 day, 2 days, 3 days, 5 days and 30 days). Gene values of around 1.5–2.0 reflect gene stability. Values up to 2.0 represent the non-stability of the genes studied. The graph in Figure 7a shows the genetic stability found for BestKeeper, which presents the *GAPDH* gene (1.356) as the most stable gene to be used as a reference gene in qPCR assays. Figure 7b shows the predicted genetic stability for BestKeeper, which presents the genes *GAPDH* (0.806), *NADH* (0.806), *COX2* (0.851) and *ATP1* (1.678) as the most stable genes to have their uses considered as reference genes. Finally, the GeNorm program found *GAPDH*, *NADH* and *ATP1* to be the most stable genes for use as reference genes in gene expression assays (Figure 7c). *GAPDH* is considered in all results as the best reference gene for gene expression analysis in blueberry plants, followed by *NADH* and *ATP1*.

## 3. Discussion

The novelty of this work is that it uses a population (clones) of colchicine-regenerated blueberries from a single donor (cv. Duke) for the validation of reference genes. Selected clones showed stability in phenotypic variability during two years (asexual generations) in a field trial under natural environmental conditions. The clones selected for this study (26 with adequate RNA quality) come from a total of 11,565 blueberry plants that were regenerated in vitro from different colchicine exposure treatments. The framework strategy of the project is to characterize stable genetic variability in natural environmental interaction. In this sense, all surviving clones were planted in the field regardless of the colchicine treatment where they originated. Any possible correlation between colchicine treatments and the efficiency of the selection of clones with commercial characteristics will be part of the future results of the project.

An ideal reference gene should have stable expression across a wide range of environmental conditions, tissues and growth stages. Therefore, to study whether colchicine treatments resulted in genomic changes affecting gene expression of specific genes, primers were designed for some housekeeping genes using genomic sequences of Vaccinium species such as *Vaccinium myrtillus* (*GAPDH*) and *Vaccinium macrocarpon* (*ATP1, NADH, RPOB* and *COX2*) as reference genomes. These genes can be constitutive in blueberry plants under normal conditions. Colchicine treatment was applied at 1 day, 2 days, 3 days, 5 days and 30 days, and the selected plants (clones) showed phenotypic variability and morphological changes in traits such as size, coloration and leaf size compared to the untreated control (Figure 2). In the plant kingdom, polyploidy is remarkably common, and polyploidization can be an important driving force in facilitating plant breeding [32]. Polyploidy has substantial phenotypic effects, such as increased cell and organ size and increased vigor and biomass, and additional phenotypic and molecular variation may arise soon after polyploid formation [32]. Colchicine treatments applied to blueberry clones at different timings may have induced different degrees of ploidy (colchiploids) as evidenced by phenotypic variability in the different traits evaluated; however, potential correlations between the time of exposure and phenotypic variability were not determined at this time. The success rate could be attributed to the species under investigation, the colchicine application protocol and the explant used [33]. However, the cell and chromosome sizes of these colchicine-treated blueberry clones still need to be assessed to demonstrate if the observed phenotypic variability could be correlated to their ploidy levels or genomic changes.

The qPCR method is one of the most sensitive and efficient methods for examining expression patterns, and together with the quantification of messenger RNA, it is widely used in gene expression studies. However, some factors may impair the quality of the qPCR technique, and the low efficiency of the primers coupled with the variability generated by colchicine treatments could increase the quality of the technique. However, *GAPDH* reference genes that were stably expressed in these plant systems were found to have low expression stability in hexaploid oat samples in both seed and endosperm [34]. Structural variation has been found to drive traits of interest in fruit and cereal crops [35], which was the target in polyploid induction. Indicative of the specificity of the primers is that they can maintain their action during serial dilutions without loss of efficiency and specificity. Primers for *ATP1*, *COX2*, *GAPDH* and *MATK* genes have shown an efficiency of around 91.09 to 109.36 and an R^2^ coefficient of 0.985 to 0.995. The primers with the best performance were those of the *MATK* generation, with their efficiency at 100.08 and R^2^ coefficient of 0.995, followed by the *COX2* gene primer, with its efficiency at 99.02 and R^2^ coefficient of 0.994. According to our study, this primer set is recommended for use as reference genes in qPCR assays using blueberry plants or plants of the genus Vaccinium, once the standard efficiency pattern is present. Dorak et al. [36] reported a comparison of efficiencies in optimized qPCR reactions for nine different *real-time PCR* kits, finding an average value of 103% (94–128%), which confirms that the efficiency obtained in this work is in the valid range for such a test. Reference genes have already been characterized for allopolyploid and autopolyploid species; however, this is the first time that they have been determined from a population of plants of the genus Vaccinium regenerated on a colchicine-supplemented medium [37,38,39,40,41]. Another study in *Arabidopsis thaliana* [42], using representative mitochondrial genes, atp1, rps4, nad6 and cox1, as targets for quantitative qPCR analysis where the selected genes represent mitochondrial chondrome-dependent functions of ATP synthesis, electron transport and gene expression, found similar results in these reference genes.

Gene expression analysis of some housekeeping genes of blueberry plants was performed using two approaches: absolute gene expression and relative gene expression between plants that underwent colchicine treatment and control plants (no colchicine treatment). These analyses were performed to understand whether colchicine treatment was able to change the polyploidy of blueberry plants and thus lead to differential expression of certain housekeeping genes. In terms of absolute expression, the *ATP1* gene showed a slight increase after 2 days of colchicine treatment. The *COX2, GAPDH* and *RPOB* genes showed an increase in absolute gene expression levels in the samples with 2 and 5 days of treatment; *MATK* and *NADH* genes showed an increase in expression in the samples with 1 day, 2 days and 5 days of colchicine treatments. *NADH* showed an expressive increase in expression after 5 days in colchicine conditions. This antimitotic altered expression in all assays compared to control samples, resulting in increased expression mainly in the 1-, 2- and 5-day treatments. The presence of colchicine may modify global gene expression when plants are exposed to colchicine for a few days. However, all genes showed low expression in the 30-day treatments. It is possible that large exposure of blueberry clones in colchicine may have an opposite effect on the absolute expression of certain genes, leading to decreased expression levels. Induction of polyploid plants of the myxoploid type is unstable and an undesirable outcome [43] as competition occurs between the original cells and the polyploid cells, leading to the elimination of the latter. 

RNA-seq analysis revealed that, in general, the transcript level showed a negative correlation, suggesting that gene methylation was copy-number-dependent or associated with different subgenomes and providing insight into the role of epigenetic variation in the evolution of the polyploid genome [44].

The relative expression in this study is not statistically significant (*p* < 0.05). However, the results were presented to improve the understanding of the gene expression behavior of the studied genes when subjected to colchicine treatment. The *GAPDH* (glyceraldehyde 3-phosphate dehydrogenase) gene was used as a reference gene. The expression of the *ATP1* gene showed a slight increase at 2 days of treatment and an expressive increase at 5 days of colchicine treatment. *COX2* showed a high expression at 3 days and a slight increase at 30 days of treatment. In addition, the vast majority of control samples showed lower expression levels for some of the other treatments, except for *COX2* and *RPOB* genes. The samples treated with colchicine for 30 days showed the lowest expression values for the genes studied, with the exception of the *COX2* gene. The *MATK* gene showed increased expression after 1 day of colchicine treatment, and the *NADH* gene showed increased expression at 3 days. Finally, the *RPOB* gene had significantly increased expression after 1 day and 2 days of treatment. Considering relative expression, colchicine treatment generally promoted increased expression at 1, 2 and 3 days. Colchicine treatment might have influenced an increase in expression of these genes on some day of treatment (perhaps due to the promotion of polyploidy), which cannot be observed in the control samples. This last strategy of polyploidy quantification is convenient for investigating physiological changes in gene expression levels, cell size, chromosomal level and morphological changes caused by polyploidy induction through colchicine, which has already been worked out and reported in the studies of Bamboo [32], Nauclea Cadamba [33] and Petunias [43].

Housekeeping genes have been used for qPCR standardization, in many cases without any experimental verification. One of the most famous cases is that of *glyceraldehyde-3-phosphate dehydrogenase*, which, after being used as a reference gene for many years, has been shown to have differential expression in different tissues or samples [45]. This has led to the consideration that a gene can be used as a reference in a qPCR analysis once it has been previously confirmed that its expression is stable in the samples to be analyzed, independently of its expression in other tissues or experimental conditions. Previous experiments have been necessary to validate the use of a gene as a reference, and new computational tools have been designed to test the stability of these genes as a reference. The genes analyzed in this study are novel and efficient for blueberry plants and other species of the genus as they are uniquely designed for them. Among the most commonly used tools are *GeNorm*, *NormFinder* and *BestKeeper* [46]. The analyses performed are shown in Figure 7, which shows that the most stable genes for these samples are *GAPDH*, *ATP1*, *NADH* and *COX2*, as they presented an M value of around 1.5 to 2.0 and have been indicated as the most stable by *BestKeeper*, *NormFinder*, *GeNorm* and the classification performed. This is the first study to define how reference genes are used in gene expression assays for blueberry plants. This set of genes could be used for gene expression studies with blueberry plants and other plants of the genus Vaccinium once the primers are designed for *Vaccinium myrtillus* and *Vaccinium macrocarpon*.

*GeNorm*, *NormFinder* and *BestKeeper* are three statistical analysis-based programs commonly used by researchers to assess the robustness of a gene to be used as a reference gene in RT-qPCR analysis. Using GeNorm, which has recently been noted as one of the best methods for determining the most stable expressed genes for qRT-PCR analysis [47,48,49]. In addition, GeNorm provides more information about an optimal number of genes in a given experimental dataset. The stability of a candidate gene is determined by pairwise comparison of the variation of the expression ratio in GeNorm to avoid co-regulation, following the lead of many other reports [21,49]. The working principle of NormFinder is similar to that of the GeNorm program, but the latter can select suitable combinations and the optimal number of reference genes. Unlike GeNorm and NormFinder, the BestKeeper software makes calculations directly using Cq values [50]. The final range of reference genes was determined with the RefFinder software, an easy-to-use web-based comprehensive tool that integrates GeNorm, NormFinder and BestKeeper [51]. Therefore, the use of multiple programs in a comprehensive analysis will help in obtaining an accurate reference gene.

## 4. Materials and Methods

### 4.1. Vegetative Material

Blueberry seedling stocks of the genotype Duke were established in vitro according to the following: Essentially, adventitious buds (~5 cm) were surface disinfected in a commercial 10% NaOCl and 0.1% Tween 20 solution (15 min) followed by a treatment in 70% ethanol (5 min) and then three rinses in sterile distilled water. Shoot tips were grown for 8 weeks in McCown’s Woody Plant (WP) medium supplemented with 30 g/L sucrose and 1 mg/L 2iP (N6-[2-isopentenyl] adenine). The culture medium was solidified with 6 g/L agar and the pH was adjusted to 5.2 before autoclaving at 121 °C for 20 min. Plant cultures were maintained at 23 °C ± 2 °C for a 16 h photoperiod under a combination of natural light and cool white fluorescent tubes at a light intensity of 60 μMm^−2^s^−1^. For in vitro micropropagation, nodal segments of in vitro seedlings were subcultured monthly in the culture medium described above.

### 4.2. Colchicine Treatments and Acclimatization

Segments of 21-day-old blueberry seedlings were subcultured on micropropagation medium containing colchicine 1 mg/L and maintained under in vitro conditions for 1 day, 2 days, 3 days, 5 days and 30 days (treatments). Each seedling regenerated from individual shoots was considered as an independent clone. Each clone was carefully individualized, washed in water and planted in 128-cell plug trays (cell volume 25 cm^3^) containing a mixture of pine compost and zeolite (2:1). The trays were kept in the greenhouse, and the relative humidity was gradually reduced: 90% → 80% → 70% at 10-day intervals under natural photoperiod. Clones that survived transplanting under greenhouse conditions were evaluated for 6 months for the following characteristics: plant vigor, leaf shape, axillary bud distribution and leaf coloring. After 1 year, the clones were planted under field conditions with their respective identifiers.

### 4.3. Data Processing in the Field Trial

A correlation study was carried out for the different characteristics evaluated in the field phase. An algorithm was constructed in R code with the software RStudio (2021.09.1+372) for the generation of cloud graphs contrasting the characteristics evaluated in all the clones regenerated in colchicine.

### 4.4. Sample Processing

For blueberries already established in field conditions, three to four leaves were collected from a total of 38 plants showing the highest contrast and agromorphological variability in the characteristics of dominant leaf coloring, number of apical leaves, number of stems, presence or absence of fruit, plant height and stem diameter (Figure 1 and Figure 2) to proceed to RNA extraction. Collected leaves were crushed in a mortar with liquid nitrogen; samples were macerated to a powder and transferred to a sterile tube containing RNA extraction lysis buffer using the Spectrum Total Plant RNA Kit (Sigma Aldrich, Louis, MO, USA). Procedures followed the manufacturer’s instructions. RNA quality was verified by 1% agarose gel electrophoresis running at 80 volts initially and then 100 volts for one hour. RNA purity and concentration were measured in a Nanodrop spectrophotometer (Thermo Fisher Scientific, Waltham, MA, USA). In total, high-quality RNA was obtained from 26 different clones. RNAs that were not visualized on the agarose gel were removed from the study.

### 4.5. cDNA Synthesis

The AffinityScript QPCR kit (Agilent, Santa Clara, CA, USA) was used to produce cDNA from blueberry plant RNA according to the manufacturer’s instructions. After measuring the purity and concentration of RNAs, each sample was diluted to a concentration of 40 ng/μL, and 5 μL (200 ng) was used for cDNA synthesis. The reaction mixture contained 10 μL of master mix (2×), 3 μL of DT oligo or random primer, 1 μL of RT/RNAsa affinity sequence and 1 μL of sterile RNAsa-free water, for a final volume 20 μL. Reactions were incubated at 25 °C for 5 min, 42 °C for 45 min and 95 °C for 5 min. The cDNAs were kept at −20 °C throughout the study.

### 4.6. Primer Design and cDNA Viability Testing

For primer design of the *GAPDH*, *ATP1*, *NADH*, *RPOB* and *COX2* genes, the PRIMER-BLAST program (https://www.ncbi.nlm.nih.gov/tools/primer-blast/, accessed on 30 July 2022) was used in its default settings. The reference genomes were *Vaccinium myrtillus* (*GAPDH*) and *Vaccinium macrocarpon* (*ATP1, NADH, RPOB* and *COX2*). All primer characteristics are summarized in Table 2.

To determine the conditions of use for each primer, PCR reactions were carried out in a final volume of 10 μL, using 1 μL of cDNA (control), 0.2 μM of each primer, 0.4 mM dNTPs, 5 mM MgCl_2_, 1X of 10X Buffer and 1.5 U of Taq-DNA polymerase enzyme. PCR products were verified by 1% agarose gel electrophoresis at 80 V for 5 min and then 100 V for one hour. After assessing the specificity of the primers in single fragment amplification, all cDNAs were assessed for viability using the *GAPDH* primer set. All 26 cDNAs were suitable for use in our study.

### 4.7. Analysis of Primer Activity and Efficiency

For the *qPCR* assays, the efficiency of the primers (90–110%) was verified with a correlation factor R^2^ equal to or greater than 0.985 and a slope between −3.1 and −3.6. To perform the standard curve and determine the efficiency of the established primers (*GAPDH*, *ATP1*, *NADH*, *RPOB* and *COX2*), control samples (triplicate) were used. To achieve these objectives, serial dilutions of 10^−1^, 10^−2^, 10^−3^, 10^−4^, 10^−5^ and 10^−6^ of preliminary *qPCR* product were performed. For subsequent analyses, the set of primers showing the above conditions was used.

### 4.8. Stability Analysis for Use as Reference Genes

To test the potential for use as reference genes in other *qPCR* assays, an analysis was performed in the RefFinder software (https://www.heartcure.com.au/reffinder/, accessed at 5 August 2022). For the analyses, the absolute Cqs of each gene separated by treatment and with all treatments were used. According to the program protocol, it uses GeNorm, NormFinder, BestKeeper and the Delta-Ct comparative method) to compare and rank the candidate reference genes. Based on the rankings of each program, it assigns an appropriate weight to an individual gene and calculates the geometric mean of their weights for the final overall ranking. Therefore, RefFinder offers the study of the stability of the reference gene expressed in an M-score, where if the value found is below 1.5–2.0, the primer is stable for use as a reference primer in gene expression assays.

### 4.9. Statistical Analysis for Gene Expression Assays

Statistical analysis was performed using Prism version 5 software. Analysis of variance for samples with non-parametric distribution (Kruskal–Wallis test) was used to detect the likely differential expression in colchicine-treated samples relative to control samples. Absolute expression was obtained by averaging the Cq (TC) of each gene for each treatment (1 day, 2 days, 3 days, 5 days and 30 days). Relative expression was obtained by Delta-Delta-CT methodology, where the expression of the *ATP1, COX2, MATK, NADH* and *RPOB* genes was assessed in comparison with control samples and the expression of the *GAPDH* gene, which was used in this analysis as a reference gene.

## 5. Conclusions

When analyzing the likely changes at the molecular level, the initial periods (1–5 days) of colchicine treatment could promote variation in gene expression of regenerated clone genes for specific genes such as *ATP1*, *COX2*, *GAPDH*, *MATK*, *NADH* and *RPOB*. However, prolonged exposure (30 days) could decrease the expression of the genes studied. 

In qPCR assays using blueberry clones with phenotypic variability, high efficiency was demonstrated using primers designed for *ATP1*, *COX2*, *GAPDH* and *MATK* genes. In addition, the *GAPDH*, *ATP1*, *NADH* and *COX2* genes showed high stability, validating their potential use as reference genes for gene expression assays in Vaccinium plants.

## Figures and Tables

**Figure 1 plants-11-02645-f001:**
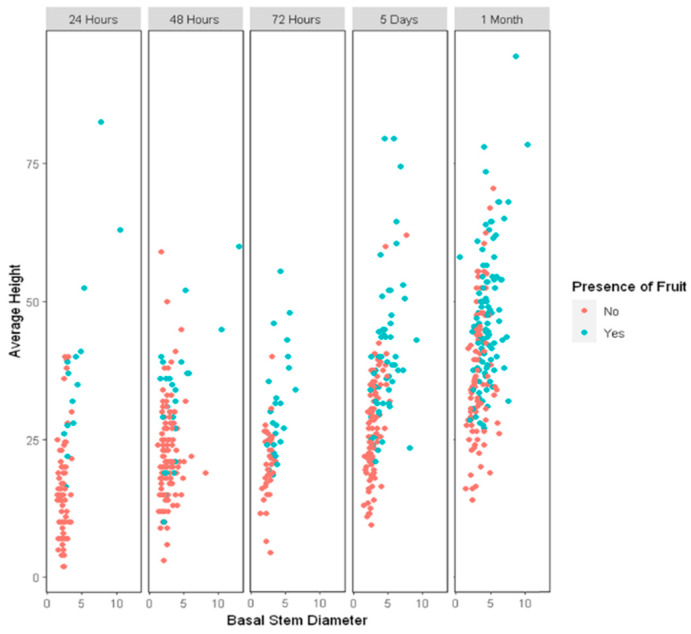
Phenotypic traits evaluated in the field study stage of blueberries (*Vaccinium corymbosum*), showing the variability between height and basal diameters in plants exposed to the same environmental conditions.

**Figure 2 plants-11-02645-f002:**
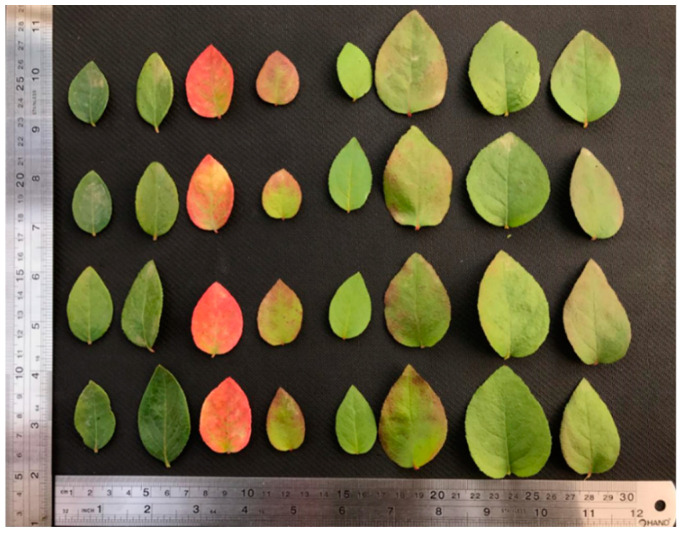
Example of genetic variability (morphology) evidence in the studied clones of blueberries regenerated in colchicine. Vertical: Replica leaves of individual clones. Horizontal: Blueberry clones under evaluation. Left: Donor genotype Duke.

**Figure 3 plants-11-02645-f003:**
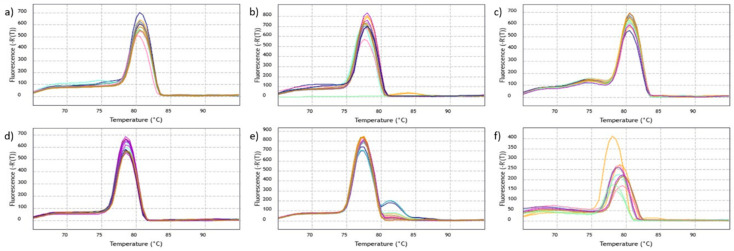
Melt curves of *ATP1, COX2, GAPDH, MATK, NADH* and *RPOB* gene primers: (**a**) *ATP1*; (**b**) *COX2*; (**c**) *GAPDH*; (**d**) *MATK*; (**e**) *NADH*; (**f**) *RPOB*.

**Figure 4 plants-11-02645-f004:**
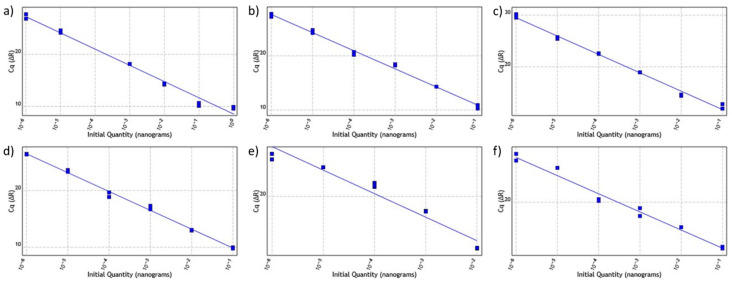
Standard curves of *ATP1, COX2, GAPDH, MATK, NADH* and *RPOB* gene primers: (**a**) *ATP1*; (**b**) *COX2*; (**c**) *GAPDH*; (**d**) *MATK*; (**e**) *NADH*; (**f**) *RPOB*.

**Figure 5 plants-11-02645-f005:**
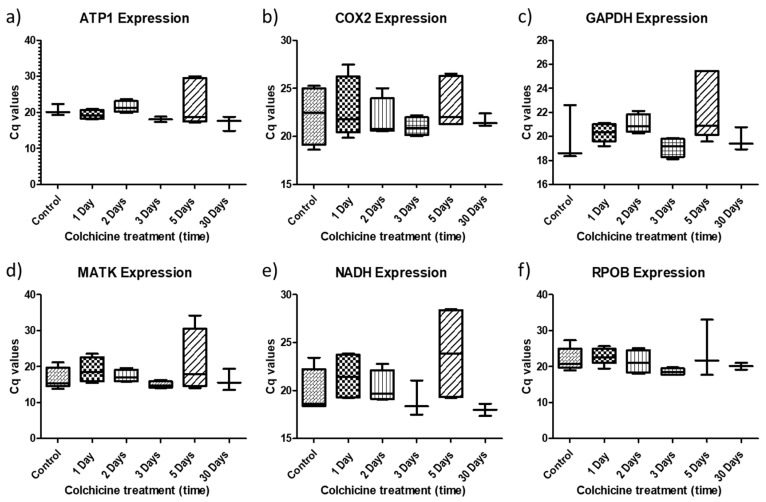
Graphs of absolute expression of *ATP1, COX2, GAPDH, MATK, NADH* and *RPOB* genes in blueberry plants subjected to different treatments with colchicine (1 day, 2 days, 3 days, 5 days and 30 days): (**a**) *ATP1*; (**b**) *COX2*; (**c**) *GAPDH*; (**d**) *MATK*; (**e**) *NADH*; (**f**) *RPOB*.

**Figure 6 plants-11-02645-f006:**
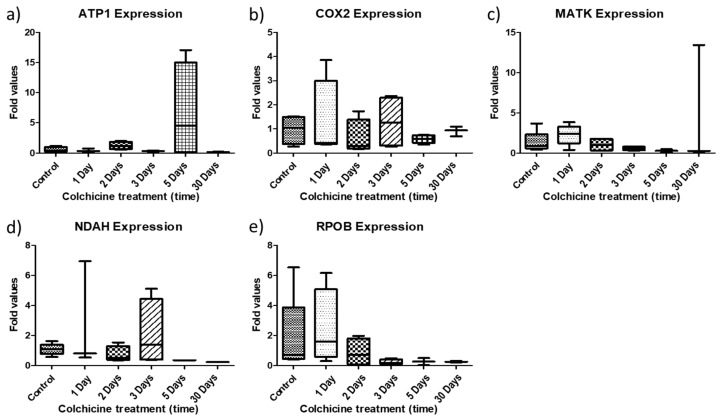
Graphs of relative expression of *ATP1, COX2, MATK, NADH* and *RPOB* genes in blueberry plants subjected to different treatments with colchicine (1 day, 2 days, 3 days, 5 days and 30 days). The *GAPDH* gene was used as the reference gene. (**a**) *ATP1*; (**b**) *COX2*; (**c**) *MATK*, (**d**) *NADH*; (**e**) *RPOB*.

**Figure 7 plants-11-02645-f007:**
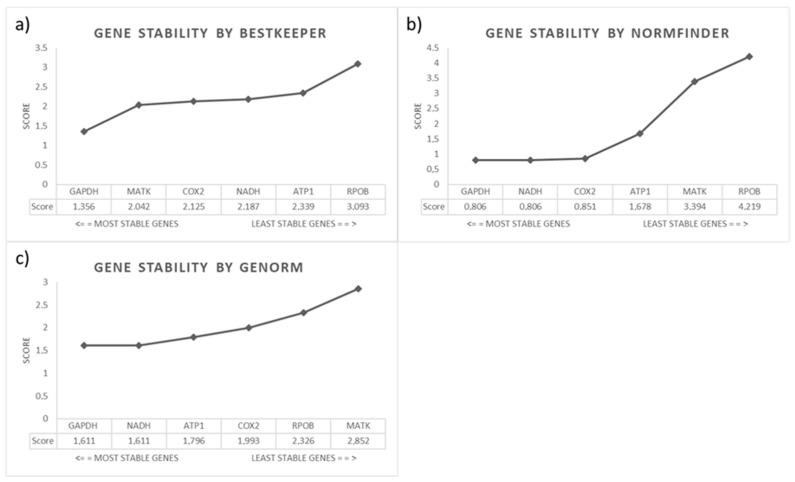
Graphs of gene stability analysis of *ATP1, COX2, GAPDH, MATK, NADH* and *RPOB* genes from blueberry plants treated with colchicine (1 day, 2 days, 3 days, 5 days and 30 days) using RefFinder software: (**a**) BestKeeper software results; (**b**) NormFinder software results; (**c**) GeNorm software results.

**Table 1 plants-11-02645-t001:** Data of efficiency and activity of each set of primers for *ATP1, COX2, GAPDH, MATK, NADH* and *RPOB* genes.

Primer	Efficiency	Slope	R^2^
*ATP1*	109.36	−3.11	0.985
*COX2*	99.02	−3.34	0.994
*GAPDH*	91.07	−3.55	0.993
*MATK*	100.08	−3.32	0.995
*NADH*	113.47	−3.036	0.954
*RPOB*	109.74	−3.10	0.980

**Table 2 plants-11-02645-t002:** Specific primers from plants of Vaccinium genus.

Gene	Sequence	Protein ID	Organism	Start	End	Primer Size (bp)	CG content (%)	Tm	Product Size
*GAPDH F*	AGGTTTGGCATCGTTGAGGG	AAM96896.1	*Vaccinium myrtillus*	134	153	20	55	60.61	174
*GAPDH R*	AAGGGCAGGCAACACCTTAC	AAM96896.1	*Vaccinium myrtillus*	307	288	20	55	60.54	174
*RPOB F*	GTTCACTCGGTTTAGCGGGG	YP_007025937.1	*Vaccinium macrocarpon*	2.6	2619	20	60	61.02	199
*RPOB R*	GGATCGCCGGTTCTTCCATC	YP_007025937.1	*Vaccinium macrocarpon*	2798	2779	20	60	60.88	199
*MATK F*	CCCTCGACTTTCTGGGCTATC	YP_007025894.1	*Vaccinium macrocarpon*	983	1003	20	57	59.93	195
*MATK R*	CCGGCTTACTCATGGGATGT	YP_007025894.1	*Vaccinium macrocarpon*	1177	1158	20	55	59.53	195
*ATP1 F*	GACCGCCTTACCCGTCATTG	YP_008999606.1	*Vaccinium macrocarpon*	972	991	20	60	60.09	181
*ATP1 R*	TTTCAACTGAGCGGCAGACC	YP_008999606.1	*Vaccinium macrocarpon*	1152	1133	20	55	60.88	181
*COX2 F*	GGGTCTTGGTTCGCGCTTTA	YP_008999605.1	*Vaccinium macrocarpon*	158	177	20	55	60.95	181
*COX2 R*	GCTGGATCGACTACTACCTCGT	YP_008999605.1	*Vaccinium macrocarpon*	338	317	20	54	61.06	181
*NADH F*	TTATTTGGGACCCGCGAGGA	YP_007025953.1	*Vaccinium macrocarpon*	1563	1582	20	55	61.27	170
*NADH R*	CGCCACAAATTCCATCCCCA	YP_007025953.1	*Vaccinium macrocarpon*	1732	1713	20	55	60.97	170

## Data Availability

Not applicable.

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
