# Peer review of "Validation of Reference Genes in a Population of Blueberry (Vaccinium corymbosum) Plants Regenerated in Colchicine"

_plants, 2022, doi:10.3390/plants11192645_

Round 1

Reviewer 1 Report

In this research, the tube seedlings of Duke blueberry were used as materials to be treared by colchicine, and then detect the expression changes of some genes to find reference genes.  The test design is good, the results is believable. The research is significant for understanding genetic background of blueberry and promote study on molecular biology about bluberry. 

line 88: one "chicory" should be deleted;

line 110: "genus" should be "species" or "plants".

Author Response

Corrections were made , Thank you 

Reviewer 2 Report

The manuscript titled Validation of reference genes in a population of blueberry (Vaccinium corymbosum) plants regenerated on colchicine” is a unique and novel study in the field of polyploidy breeding. However, minor changes are needed to make it acceptable for the scientific community

Introduction:

Line 49: What is meant by abiotic and biotic traits? Explain them in detail

Results:

Based on the results, the ploidy level of clones was not quantified, and reference gene expression was studied by random selection of phenotypic variable clones. However, according to my experience, there are no reliable results as, during colchicine treatment, the production of mixoploids/chimeras is a common phenomenon (mentioned in lines 312-313). These plants were unstable and didn’t yield authentic results regarding reference gene expression. Thus, the authors must provide an explanation regarding this concern.

Figure 2: Leaf variation representation is from 38 selected clones or from the overall colchicine-treated population? Mentioned it in the caption

Discussion:

Discussion related to the increase and decrease of gene expression at different days of treatment needs further explanation. Incorporate metabolic processes involved in gene expression with respect to colchicine concentration or exposure.

Methodology:

Why there is a drastic change in treatments in terms of colchicine treatment from 1,2,3 and 5 days to abruptly 30 days of continuous treatment? As it is already clear from various works of literature that at high concentration or exposure time colchicine toxicity leads to plant death. Therefore, it is better to study at an intermediate level rather than choosing treatments of very low and very high time duration. The authors must clarify their treatments to avoid confusion. 

Author Response

please find file attached 
